# Ancient Feminine Archetypes in Shi'i Islam

Amina Inloes 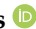

Department of Islamic Studies, The Islamic College, London NW10 2SW, UK; a.inloes@islamic-college.ac.uk

**Abstract:** This paper explores archetypes of femininity associated with Fāṭimah al-Zahrāʾ in Twelver Shi'i hagiography through consideration of a broad range of archetypes found in the study of narrative and mythology. Many archetypes associated with goddesses of antiquity recur in portrayals of Fāṭimah al-Zahrāʾ, suggesting either cultural influence or universal archetypes. For instance, Fāṭimah embodies a youthful, innocent, virginal goddess; Jung's light and dark mother figure; and the lamenting goddess. Similar archetypes are projected onto other sacred women in Shi'ism, such as Zaynab bint ʿAlī and Fāṭimah al-Maʿṣūmah. However, other feminine archetypes are absent, some are sublimated onto male figures, and some are banalized through translating the esoteric into the exoteric. This leaves gaps in the narrative models available to faithful women. Furthermore, embodying archetypes like lamenting and suffering may be undesirable. While reformist portrayals of Fāṭimah have attempted to present her as a model for female activism, historical and hagiographical archetypes of Fāṭimah inherently clash and are difficult to disentangle. Nonetheless, considering how hagiography differs from history can help understand how the mythic does not always translate well to the mundane.

**Keywords:** Islam; Shi'ism; Karbala; Fāṭimah al-Zahrāʾ; ʿAbbās ibn ʿAlī; ʿAlī ibn Abī Ṭālib; Shi'i *mythos*

## 1. Introduction

> There are many who would say that we are emerging today from the artificial polarities of the male God religions in search within ourselves and our world of the ecological wholeness of Goddess, who contains and celebrates light *and* dark, life *and* death, male *and* female, and whose source is the inner depths rather than the airy heights. —*Goddess: Myths of the Female Divine* (Leeming and Page 1994, p. 3)

Archetypes may seem antithetical to lived experience. Nevertheless, gender archetypes in narrative guide people to what is desirable or even possible; they mark out signposts and boundaries for the game of life. This paper explores how ancient archetypes of femininity associated with goddesses in and around the Middle East and North Africa persist in in Twelver Shi'i sacred narrative[1] through the portrayal of Fāṭimah al-Zahrāʾ, the woman most central to Shi'i hagiography whom faithful women are urged to emulate. While many goddess archetypes recur in portrayals of Fāṭimah al-Zahrāʾ, others are sublimated onto men or absent. This, in turn, reduces the scope of archetypes available to women in Shi'i sacred narrative. Furthermore, other archetypes—such as hiddenness—occur in portrayals of Fāṭimah al-Zahrāʾ, but devolved from esoteric to exoteric understandings, resulting in, at best, banalization, and, at worst, the marginalization of women. Understanding these archetypes in light of their ancient expressions can help uncover more constructive paths to expressing them in daily life. Additionally, like other studies on the feminine in Shi'ism, Sufism, and esoteric Islam, exploring archetypes of feminine divinity surrounding Fāṭimah al-Zahrāʾ helps challenge the presumption that the divine feminine is absent from the Abrahamic traditions.[2]

This paper takes a broad definition of archetype, as used in contemporary studies on secular and sacred narrative, and treats archetypes as:

primordial images, myths, and evolutionary symbols that represent [possibly] inborn and [probably] universal ways of perceiving and comprehending the world [...] [that] unite humankind through symbols that provide individuals with wisdom about the past and predispose people to experience the world as their ancestors did. [...] Archetypal heroes that appear in these stories serve as role models and help individuals expand their emotional repertoires. [...] They are seen as symbols that help people overcome adversity, reveal prescriptions for change, and encourage ordinary individuals to access the 'hero within'. (Enns 1994, pp. 127–33)

[Additions in brackets mine.]

This paper is not attempting to draw conclusions about archetypes—for instance, whether they are truly inborn or universal. Rather, archetypes are being used to better understand how Shi'i sacred narrative influences Shi'i women's lived experiences. Primarily, it addresses pre-modern/pre-reformist portrayals of Fāṭimah, although reformist portrayals are also acknowledged. Specific groups of archetypes explored in this paper include:

- Jung's **mother goddess** archetype, which falls into **light** (good, benevolent), **neutral** (no particular moral alignment), and **dark** (evil, frightening) models (Jung 1968, pp. 75–111).
- The categories of the **warrior**, **magician**, **orphan**, **innocent**, **wanderer**, and **martyr**, potentially applicable to both genders (Pearson 1986, 1991).
- The categories of **virgin goddesses**, **vulnerable goddesses** (including the lamenting or suffering goddess), and a **transformative goddess** (Bollen 1984).
- The division of womanhood into **maiden**, **mother**, and **crone** phases.
- The goddess as the personification of **justice**.

While many, or all, of the above ideas may be critiqued, and the list is by no means comprehensive, it nonetheless provides a sufficient set of concepts to facilitate a discussion of feminine archetypes in Shi'i sacred narrative.

It should be noted here that many Muslims, including Shi'is, balk at examining Islam through the lens of pre-Islamic custom; rather, Islam is divided from *jāhiliyyah* by an invisible line of time. Contemporary Shi'is often argue vigorously that Shi'i belief and practice derive directly from the Prophet. This is not only for the sake of apologetics— proving that Shi'ism is the 'real' Islam—but also because sectarian violence against Shi'is (such as mosque bombings) is goaded on by the accusation that Shi'ism is less 'authentic' than Sunnism. The goal of this paper is not to argue that Shi'ism is any more or less authentic than Sunnism; archetypes, by definition, recur throughout cultures and religions, including both Sunnism and Shi'ism. Furthermore, key aspects of Islam, such as the hajj, predate the Prophet, and so there is ample reason to discuss the ancient heritage. Second, the subject of goddesses can be uncomfortable to some Muslims: not only does the Qur'an condemn three pre-Islamic goddesses (Lāt, ʿUzzah, and Manāt, as mentioned in Qur'an 53:19–20), but many a Muslim has been 'gifted' a pamphlet informing them that Allah is really Al-Lāt, the Moon God. However, the Qur'an itself is condemning idolatry, not mythical archetypes of femininity. Lastly, I am not intending this paper to be prescriptive. Just because Fāṭimah has been understood archetypally does not mean that faithful Shi'a women must attempt to emulate those archetypes. Rather, I am advocating the opposite: real women, like the historical Fāṭimah, are complex, and do not merely incarnate as archetypes, emerging from the depths of the subconscious or Plato's World of Forms. While it is natural that Shi'i hagiography would fall into the comfortable model of time-tested, cross-cultural archetypes, living real life as an archetype is impractical.

(Note: Throughout this paper, keywords pertaining to archetypes or imagery have been **boldfaced**.)

## 2. Archetypes of Femininity in Twelver Shi'ism

> Just as Goddess can appear anywhere at any time, she can take form as the Child-Maiden, the Mother, or the Crone [...] Goddess can be many things [...] She can be the universe itself, the source of all being, the sacred temple, the divine child, the Holy Virgin, the Earth Mother-nurturer, the madly hysterical destroyer, the femme fatale, the consort or mother of God, the willing participant in the castration or murder of her husband, even the lover of her son. (Leeming and Page 1994, p. 4)

In Twelver Shi'ism, the primary feminine archetype is Fāṭimah al-Zahrāʾ, and the primary masculine archetype is ʿAlī ibn Abī Ṭālib. Masculinity and femininity are traditionally treated as separate, similar to the ideas of yin and yang or the animus and anima. Some scholars, such as Ayatollah Javādī Āmolī in his *Zan dar Āyīneh-ye Jalāl va Jamāl* (Women in the Mirror of Majesty and Beauty), elevate these differences to a theological level—describing masculine (*jamālī*) and feminine (*jalālī*) attributes of God, such as 'the King' and 'the Vengeful' versus 'the Loving' and 'the Compassionate' (Murata 1992, p. 69). Others present them as inborn natural tendencies in biological males and females (Inloes 2019, pp. 214–7). While contemporary writing on archetypes has—in my view, rightly—criticized the underlying assumption of gender dualism in masculine and feminine archetypes, nonetheless, Muslim polemics about gender roles center on this assumption, even if ḥadīth and jurisprudence can be read in a more nuanced manner (Inloes 2019, pp. 214–7). Both ʿAlī and Fāṭimah are historical figures: ʿAlī (d. 661) was the first Shiʿi Imam and the fourth caliph, while his wife Fāṭimah (d. 632) was the daughter of the Prophet Muḥammad; the remaining eleven Shiʿi Imams were descended from them. Nevertheless, history and hagiography blur; most Shiʿis treat hagiography as historical fact.

The spiritual status accorded to Fāṭimah al-Zahrāʾ in Shiʿism has been amply discussed elsewhere. As Henry (Henri) Corbin describes her:

> She is the *supracelestial* Earth [...] the manifested Form [...] The very soul (*nafs*, *Anima*) of the Imams; she is the Threshold (*bāb*) through which the Imams effuse the gift of their light [...]. (Corbin 1977, pp. 63–64, cited in Warner 2011, pp. 147–62)

In an Iranian context, Corbin understands this divine aspect of Fāṭimah to be a way that Iranian culture and religion preserved a sense of the divine feminine after the adoption of Abrahamic monotheism, and likens it to the Jungian concept of **Sophia** as the personification of feminine wisdom:

> [...] a tragedy in Iranian thought and consciousness. Persia became Muslim in the course of the first centuries [...]. Let us consider the hypothesis of a Sophianic consciousness that suddenly is turned upside down and falls prey to the Yahweh of Job. This tragedy has perhaps not yet been properly formulated to our consciousness. And yet, there may well have been also something like an Iranian voice that could give an "answer to Job". This response is graven in the devotion that developed in the form of Shi'ite Islam. Not only do the Holy Imams form a chain of helpful intermediate beings, but especially Fatimah, daughter of the Prophet and Mother of the Holy Imams assumes a role, just as much in the popular piety as in the theosophical speculations of Shi 'ism, and especially in Ismaeli Shi'ism, a role that makes of her person a recurrence of the patronage of Sophia. (Corbin 2019, p. 141)

Since Iran only became a majority Shi'i country in the 16th-17th century, and Shi'ism developed primarily outside of Iran, his observation does not reflect Shi'ism globally and historically in its entirety; rather, it is more pertinent to regional studies. Corbin's sense of Shi'i sacred narrative as essentially Iranian could also be countered by Ayatollah Murtaḍā Muṭahharī's stance that the fictional elements of Shi'i sacred narrative (such as fanciful stories about the Battle of Karbala) came from Najaf, the historical centre of Shi'i

scholarship, not Iran (Muṭahharī 1985, Part 1). Some might find his notion of an 'Iranian Islam' (as in his famous work *En Islam iranien*), founded on a timeless Persian culture, to be unnecessarily essentialist, or overly racialized. Here, Corbin is negative towards Islam in a way that similar thinkers, such as Seyyed Hossein Nasr, are not. Furthermore, Corbin's writings are more grounded in Ismailism rather than Twelver Shi'ism. For these reasons, Corbin's writings cannot be applied to Twelver Shi'ism uncritically (as sometimes happens). Nonetheless, Corbin does effectively put into words the spiritual embodiment of Sophia through Fāṭimah, as well as the survival of ancient ideas in newer forms.

In short, Fāṭimah is central to creation; a Shi'i *ḥadīth* says, 'Were it not for Fāṭimah, God would not have created the universe' (Shahrūdi 1405 AH, vol. 3, pp. 168–9). The name 'Fāṭimah' is linked to the divine attribute *fāṭir*, meaning 'Maker' (Amir-Moezzi 1994, p. 30). She, the Prophet, and ʿAlī figure into accounts of the heavens in the beforetime, and, according to one strand of Shi'i narrations, Eve ate from the forbidden tree because she was jealous of Fāṭimah (Inloes 2019, pp. 95–98). In this world, Fāṭimah was inerrant, spoke from the womb, and was the locus of miracles (Amir-Moezzi 1994, pp. 30–31; Clohessy 2009, pp. 102–4). As the ancestress of the future Imams, she is the isthmus between them and the Prophet just as they are the isthmus between humanity and God. After her father, the Prophet Muḥammad (d. 632) passed away, she was overwhelmed with grief; to console her, the angel Jibrāʾīl dictated to her a secret book known as *Muṣḥaf Fāṭimah* (Amir-Moezzi 1994, p. 74). Unlike the Qur'an, which is a public book and contains, among other things, legal rulings, *Muṣḥaf Fāṭimah* is private and contains esoteric knowledge; like the archetypal Fāṭimah, it is secret, spiritual, and hidden, rather than exoteric, legal, and known. In the next world, she is an **intercessor** and will demand **vengeance** against her children's enemies.

Fāṭimah, in the Shi'i view, like the Prophet Muhammad and the Imams, is **sinless**. In a study on Fāṭimah al-Zahrāʾ, George William Warner argues her sinlessness is what transforms her history into sacred history:

> A Shi'a account of Fatimah's life is a sacred account because it does not simply tell the believing reader what happened and what did not, but tells them too what should have happened and what should not [...] it is understood by the Shi'a reader that in a narrative context whatever Fatimah does will be absolutely morally correct, and any action which deliberately violates her interests will be morally corrupt. (Warner 2011, p. 148)

This excerpt from 'May Fatima Gather Our Tears' summarises Twelver Shi'i conceptions of Fāṭimah:

> Fāṭimah al-Zahrāʾ, the daughter of the prophet Muhammad, wife of the first Shi'i imam, ʿAlī, and the mother of two martyred imams, Hassan and Husayn, is revered as one of the most holy of Muslim women—sinless, spiritually perfected, and most commonly portrayed by her scholarly interpreters as the lady of sorrows and unstinting patience. Fāṭimah possesses two roles beyond being the witness to the grief expressed by the loyal followers of the Prophet Muhammad's family (Ahl-e Bait): she is a transcendent figure, created before Creation, whose eschatological role is ultimate intercessory authority on the Day of Judgment, and she is venerated for her humanity, with its attendant feelings, emotions, and desires, through which her devotees cultivate feelings of love and respect. Persian and Urdu theological and hagiographical texts from the tenth century and continuing to the present day portray Fāṭimah as a transcendent figure, whose generative light is the source of prophecy and the imamate, illuminating the heavens on the Day of Judgment. There is also a rich hagiographical tradition that portrays Fāṭimah in her very human roles of mother, daughter, and wife and as a woman whose material and emotional needs resonate with Shi'i devotees of the prophet Muhammad's family. (Ruffle 2010, p. 8)

In short, although Fāṭimah is not the Creator, she nonetheless is as close to **the source of all being** as is possible for a human within Islamic theology. While both Sunnis and Shi'is revere her, the transcendental and eschatological flavour of her hagiography is most pronounced in Shi'ism.

Like the Virgin Mary, Fāṭimah is an archetypal **mother figure**. On a mundane level, she is seen as an ideal mother and titled the 'mother of her father' (*um abīhā*). 'Mother of her father' is understood conventionally to mean that, even as a child, Fāṭimah cared for her father in times of distress, in the stead of his own mother, who had passed away in his childhood. (Whether or not this should be praised is a different matter, since some might argue that it is unhealthy for a parent to treat a child as a substitute parent. However, since Fāṭimah was born with preternatural wisdom and maturity, we can overlook this hagiographically.) However, Annemarie Schimmel also takes the expression 'the mother of her father' as a paradox, like a Zen koan, that reflects the 'divine now'; an eternal, sacred time outside the norms of conventional time, the 'still point that contains in itself all movement'—the aboriginal dreamtime that sacred figures such as Fāṭimah inhabit or even create to allow us to interact with them (Schimmel 1994, p. 117).[3]

Fāṭimah is also a spiritual **mother** whose embrace extends to her followers eternally. Jung divides the archetype of the **mother goddess** into three expressions: light (good, benevolent), neutral (no particular moral alignment), and dark (evil, frightening). Largely, Fāṭimah is associated with the **positive mother archetype**, symbolized by 'our longing with redemption, such as Paradise, the Kingdom of God, the Heavenly Jerusalem', which her intercession leads her followers to. She—like Jung's positive mother archetype—is characterized by 'maternal solicitude and sympathy; the magic authority of the female; the wisdom and spiritual exaltation that transcend reason; any helpful instinct or impulse; all that is benign; all that cherishes and sustains, that fosters growth and fertility' (Jung 1968, pp. 81–82). Nonetheless, Fāṭimah retains darker associations of Jung's mother archetype, such as **death**, the **grave**, and rebirth through **martyrdom**.

However, Fāṭimah lacks the moral ambiguity of fate goddesses such as Manāt or the devouring sexuality of Lilith;[4] she expresses light and darkness, but not the space in between. While absent from popular hagiography, the Lilith archetype can be found in the obscure figure of ʿAnāq in Twelver Shi'i *ḥadīth*, although most Twelver Shi'is have never heard of ʿAnāq (Coulon 2019; Inloes 2019, p. 113). Nevertheless, the **evil temptress** archetype, although not embodied by characters in Shi'i sacred narrative, constitutes the unexpressed **shadow** side of the woman; because this potential is latent, and potentially transgressive, it must be subdued. It is also hinted at in criticism of ʿĀʾishah, who rode out to battle against ʿAlī—who, in the Shi'i view, was appointed as an authority by God—and who is therefore tacitly criticized for being independent rather than obedient (Inloes 2015), although most Shi'is refrain from sexual innuendo about ʿĀʾishah.

Like the Virgin Mary, Fāṭimah is the eternally **lamenting**, as Annemarie Schimmel (2003, p. 30) calls her, a *mater dolorosa*. Antiquity is replete with weeping goddesses, such as Tanit (described as *Pane Baal* or 'the weeper of Baal') (Lipiński 1995), Demeter lamenting Persephone, and the lamentation for Tammuz. According to Twelver Shi'i tradition, Fāṭimah began lamenting the **martyrdom** of her son, al-Ḥusayn, upon his birth, and she will continue to lament until the Resurrection. Therefore, she is present at all gatherings of ritual lamentation (for instance, Muḥarram and ʿĀshūrāʾ commemorations), and it is popularly said that she collects tissues used to dab tears during these ceremonies. Her grief intensified towards the end of her life; after the Prophet passed away, she mourned so copiously that a special building was made for her—the *bayt al-aḥzān* ('house of sorrows')—so that her weeping would not distress the people of Medina. Through her grief, she therefore also represents the **orphan** archetype which helps the individual get in touch with abandonment and betrayal.

Twelver Shi'is also almost unilaterally hold that Fāṭimah was martyred. In contrast, most Sunnis say that, after the death of her father, she died of grief. At that point, in the Sunni narrative, her grief ends when she dies and reunites with her father in Paradise,

whereas, in the Shi'i view, she subsists forever as a martyr. The account of her martyrdom, as outlined in *Kitāb Sulaym ibn Qays* (attributed to Sulaym ibn Qays, d. 678), is as follows.[5] Shortly after the death of the Prophet, during the struggle over the caliphate, the second caliph brought a band of thugs to the house of 'Alī and Fāṭimah to force 'Alī to pay allegiance to the new caliph; they threatened to burn down their house if he did not comply. When Fāṭimah opened the door, they crushed her against it, fatally injuring her and killing her unborn child Muḥsin. She therefore not only symbolizes **birth** but also **death** and **stillbirth**. Iconic Shi'i drawings depict the men brandishing torches outside her house, in the dark of night, while Fāṭimah, wrapped in a dark veil, is doubled over—a literal portrayal of the **dark mother archetype**. Similarly, during the Battle of Karbala, just as her son, al-Ḥusayn—the archetypal martyr of Shi'ism—is slain, so too is her infant grandson, 'Abd Allāh ibn al-Ḥusayn, slain on the battle field; his blood is cast to the heavens and not a drop returns, symbolizing a **divine sacrifice**.[6]

In today's world, with the decline of infant and maternal mortality, pregnancy is often idealized as the spring of new life. However, the image of the goddess bringing both life and death—birth and stillbirth—is common throughout the ancient world, including in the Carthaginian goddess Tanit, at whose ritual sites infant burials have been found, which have led to debate over whether people merely laid their infants to rest near her, or whether she was offered infant **sacrifices** (Xella et al. 2013). The archetypal Fāṭimah seems to reflect this stark reality.

The question has been posited: 'Why, one might ask, the emphasis on sadness? Why do we hear the sobs of Fāṭimah echoing across time rather than her laughter?' (Warner 2011, p. 159) Mid-to-late twentieth-century reformers (for instance, Shahīd Bāqir al-Ṣadr, 'Alī Sharī'atī, and Imām Khumaynī) portray her grief as a political stance: Fāṭimah protested the awarding of the caliphate to someone other than 'Alī through vociferous grief, thereby modelling political activism for women. However, there is more happening on a spiritual and cosmological level. Fāṭimah's lamentation is salvific:

> Devotion to Fāṭimah's mystical and intercessory qualities orients Shi'i ritual toward an eschatological future-present, in which Fāṭimah will be transformed from the gatherer of the mourners' tears into the supreme intercessory authority on the Day of Judgment.[7]

The **redemptive** quality of lamenting Fāṭimah's suffering forms the backbone of the controversial Shi'i-made movie *Lady of Heaven* (released in 2021) in which hearing about Fāṭimah's martyrdom enables a child to talk down a would-be suicide bomber.[8] Fāṭimah's suffering is also archetypal; she is mourning not just the loss of her offspring but also the sufferings of each and every one of the Shi'a throughout time:

> Fāṭimah's function, conversely, is precisely to suffer, to suffer visibly and paradigmatically so as to illustrate the horrific reality of this injustice as its archetypal victim. While 'Alī is heroically active, Fāṭimah must remain tragically passive. [...]

> The community receives the perfection that is the Imams but Fāṭimah receive the imperfection of the community, an imperfection stretching across time to encompass all of the tyrannical ignorance which Shi'a sees in centuries of political and religious subjugation. All of this is inflicted on Fāṭimah; and so Fāṭimah weeps. (Warner 2011, pp. 154–60)

Therefore, Fāṭimah continues to weep until the return of the Mahdī, or until the Resurrection, at which time the martyrdoms will be avenged and the wrongs will be redressed. Nonetheless, there is still a more fundamental reason why Fāṭimah laments—it is because, as among many goddesses of antiquity, lamenting is an expression of feminine divinity.

Twelver Shi'i ritual lamentation ceremonies resemble the ancient Greek and Mesopotamian customs of ritual lament—even if, today, many Shi'is are careful to root them in the practice of the Prophet and his family. Ancient Greek tragedy and Shi'i ritual storytelling and

theatre also parallel each other, even if Greek tragedies were omitted from the early Graeco-Arabic translation movement. In an extensive study on the ancient Mesopotamian lament tradition, Paul Delnero observes similarities between ancient Mesopotamian and Shi'i lamentation ceremonies, but also a difference in function: while ancient Mesopotamian lament had an apotropaic purpose—to ward off calamity—Shi'i ritual lament is primarily aimed at generating grief:

> When rituals such as these are performed effectively, the grief they are capable of generating can be no less real and powerful than the grief that is felt by those who have experienced tragedy and loss more immediately and directly. (Delnero 2020, p. 3).[9]

However, while Shi'i scholars do not present the primary purpose of ritual lamentation as apotropaic, many Shi'is privately participate in ritual lamentation to seek intercession against evils, or have the lingering suspicion that if they were to neglect observances such as ʿĀshūrāʾ, something bad might happen. Shi'i ritual imagery also includes apotropaic inscriptions (Kemnitz 2022, pp. 135–8). Therefore, an **apotropaic** element still underlies Shi'i ritual lamentation. Additionally, Shi'is commonly expect prayers to be answered, or even miracles, at these ceremonies, especially during the height of the lament. From the perspective of the study of religion, ritual lamentation can be seen as one of the many ways in world religions in which emotional energy lifts the devotee out of a mundane frame of mind and fuels fervent prayer; a corollary within Islam would be celebratory Sufi practices intended to cultivate a sense of divine ecstasy. That is, lamentation opens the door to the divine.

Men in Shi'i sacred narrative also suffer and lament; all of the Imams are martyrs. However, a broader scope of archetypes is available to men; not only do they suffer and lament, but they are also warriors, statesmen, and scholars. In contrast, women primarily lament. While this archetype can offer solace and strength in times of difficulty, it can also promote passivity or socialize women to accept suffering rather than stand against it, including within the domestic sphere. Eternal lament is not an archetype which is healthy to embody. Understanding the link between lament and feminine notions of divinity in the tradition can help divide what is archetypal from what is desirable in the real world.

The narrative of Fāṭimah's martyrdom also reflects a variant of the **damsel in distress** literary trope, which traces as far back as ancient Greek tragedies involving young women, gods, and demigods; it is inherently mythic. Setting aside any negative connotations this expression may have today, key aspects of this model for our discussion here are:

> a young and presumably innocent woman is held captive against her will by an evildoer, or cannot free herself from a curse or some other psychological captivity. [...]
>
> One of the essential parts of the scenario is the heroic effort to rescue her. [...] The moral of many of these stories is not strictly about the rescuing process, but the altruistic reasons for the rescue. Evil must be defeated before the damsel can be released from its grip, and only the most heroic and purest of heart would have the power to succeed. (Pollick 2023)

In Twelver Shi'ism, Fāṭimah as portrayed as young, frail, and innocent. She is guileless and sheltered in her home. (This is a hagiographical not historical view.) In Shi'i hagiography, 'evil' connotes the enemies of the Imamate, who, literally, trap her in her home and, symbolically, trap her in grief. Thereby, she embodies Jung's feminine **injured innocent**, **virgin goddess archetype** and **vulnerable goddess archetype**. Unable to defend herself—she is not a **warrioress**—she needs a **hero**—ʿAlī—to rescue her. Sometimes, Shi'is stop here and ask why ʿAlī—a formidable warrior—did not just get up and immediately fend off the men attacking his wife, rather than doing so after a delay. According to *Kitab Sulaym ibn Qays*, ʿAlī had sworn to the Prophet that he would not fight; the earliness of this source suggests how glaring this narrative dissonance is (Qays 1386 AH, no. 4). However, a deeper reason can be proffered: before rescuing the damsel, the hero must first overcome the evil.

Since ʿAlī does not overcome the evil—that is, the family of the Prophet remain stripped of their right to authority—he cannot rescue her. Therefore, she dies, and the evil continues until the time of the Mahdī—the saviour awaited at the end times—who will restore peace, justice, and the rule of the Imams—or the Resurrection, at which time God will rule in her favour.

One could pause here and ask why Fāṭimah should not be a warrior. During the time of the Prophet, some women did go to battle. However, Shi'i hagiography emphasizes that women are non-combatants. For instance, in the account of the Battle of Karbala, one woman does seize a tentpole and move to fight back, and Imam Ḥusayn tells her to sit down, because jihad is not for women. Anecdotally, I have heard some Shi'a today ask why women such as Zaynab bint ʿAlī did not try to physically defend themselves and their family, because they would have.

There seem to be a few factors involved here. First, characterizing women as weak and sheltered better reflects ideals towards women which became popularized after the expansion of the Arab-Muslim Empire into ancient Mesopotamia. Second, it also reflects the religio-cultural emphasis on women not being seen or touched by unrelated men; on a practical level, it could be intended to discourage assault during warfare. Third, however, the Kharijite movement, which emerged during the first civil war (656–661), encouraged women to participate in jihad. Because the Kharijite movement opposed ʿAlī ibn Abī Ṭālib and his followers, it was a clear enemy to the Shi'a; therefore, rejecting female fighters may have been part of drawing a dividing line between Shi'a and Kharijites. That is, it may have been a sectarian issue. More archetypally, over the course of human history, once-powerful goddesses were reborn in the **goddess abused** archetype—such as the many goddesses mistreated by Zeus—in which the goddess, although still important, must now resort to 'begging or guile' before men (Leeming and Page 1994, p. 91).

In that regard, Fāṭimah embodies being the victim of perpetual **injustice**. This is in contrast to her husband, ʿAlī, who, although also a martyr, is usually associated with **justice**. For instance, a famous book about him is titled *The Voice of Human Justice*; he occupies the seat of a judge; and he is idealized as having the 'just' balance of ethical faculties (an Aristotelean concept influential in Islamic ethical theory). This departs from the archetype of the goddess of justice, such as Athena or the Egyptian personification of the 'goddess of justice', Maat, which leaves a starkly different message: while 'the Ancient Egyptian lived in the unshakable faith that Ma-a-t [...] was, despite periods of chaos, injustice and immorality, absolute and eternal' (Mancini 2007), the injustice meted out to Fāṭimah presages injustice for all time, telling us that the world is essentially unjust.

Fāṭimah's **youth** at the time of her martyrdom is part of the pathos; one cannot help but sympathize with a young, frail mother attacked by a band of powerful men. Possibly, this is why Shi'a say that she was 19 when she died, whereas Sunnis usually say she was around 29 (Ibn Saʿd 1988, vol. 9, p. 128; Clohessy 2009, pp. 21–31). Youthfulness emphasizes her innocence and beauty; Fāṭimah never had grey hairs. According to Shi'i *ḥadīth*, Fāṭimah was radiantly beautiful; Sunni tradition sometimes demurs. Therefore, she shares traits with the goddess who is **young**, **beautiful**, and **virginal**; although married and a mother, two of Fāṭimah's epithets are *al-ʿadhrā'* and *al-batūl*, both of which literally mean 'virgin', although these are also taken to mean that Fāṭimah did not experience a menstrual cycle, and some might interpret them to mean 'chaste'. Virgin goddesses in antiquity abound—or, more strictly, virginal goddesses who are not actually virgins—which has led to the view that the Virgin Mary was easily accepted due to the cultural precedent (See Rubin 2009). This trait can be summed up by a description of Tanit, one of the many **virgin** goddesses of antiquity who nonetheless had a consort; she was 'young and attractive (Virgin, but not *intacta*)' (Hvidberg-Hansen 1986, p. 176).

(Addressing sacred figures through epithets is also found in the *Orphic Hymns*, where Greek gods and goddesses are addressed by epithets, and thus appears to reflect an older regional practice.)

How Fāṭimah managed to be an unmenstruating married virgin mother has not been unperplexing. One explanation is that, according to a *ḥadīth* from the Prophet, Fāṭimah was a human *ḥūrī* from heaven, in the sense that she was born from a light created in Paradise (Ṣadūq 1937 AH, p. 396). (*Ḥūrīs* are beautiful maidens available to the faithful in heaven, as per Qur'an 56: 35–36.) The exegete Ṭabrisī (1995), in his exegesis on 56:35–36, explains that while the inhabitants of heaven may marry a *ḥūrī*, 'whenever their husbands come near them, they find them virgins'. Bringing the esoteric down to the exoteric level, their virginality is perpetually renewed, and so is hers. At the very least, they and she are perpetually youthful.

However, this leaves a profound gap for the faithful woman: since Fāṭimah dies at the age of 19, there is no model for the rest of womanhood. While the division of womanhood into the model of maiden, mother, and crone is rightly criticized as reducing women to their reproductive timeline, nonetheless, Fāṭimah is a **maiden mother**, and certainly does not leave any space for the **wise woman** or **crone**, whose life experience benefits the tribe. In some cultures today, such as American culture, youth for women is inordinately glorified; this makes it all the more pressing to consider the place of elderly women in religious narrative.

The role of the matriarch could have been filled in by Fāṭimah's daughter, Zaynab bint ʿAlī, who is central to the narrative of Battle of Karbala. However, although Zaynab is in her 50s during the Battle of Karbala, she is nonetheless treated as young; for instance, elegies mourn the loss of her male guardian and how, earlier, she was orphaned. Although she looks after the survivors of the Battle of Karbala, she is only a stand-in for the Imam, who is ill; she has that responsibility thrust upon her. Those who listen to the account without knowing the history might not recognize that she is, in fact, an old woman. While these narrative tropes reflect the reality of the human condition—that experiencing one loss can revive all other past losses—they keep Zaynab from being a matriarch taking power in her own right. Another revered descendant of Fāṭimah al-Zahrāʾ, Fāṭimah al-Maʿṣūmah (d. 816), who died at the age of 26, is also remembered as a virgin.

Furthermore, although such matters are hinted at discreetly, Fāṭimah is celebrated as a married **virgin** with an ideal nuptial life; ʿAlī is a perfect husband in all ways. Most women have a more challenging time navigating sexuality, especially given the vagaries of midlife, let alone the immense social changes of modernity. However, the only message available from the portrayal of Fāṭimah is that women marry and ought to marry young; everything else works itself out. Fāṭimah also epitomizes modesty, presented as a primary value for women, even though, in practice, the *ḥadīth* tradition talks about modesty equally or even more so for men, not just women. In contrast, other archetypes, such as the **fertility goddess**, **temptress**, or **initiatrix** provide a more robust library of models through which women might understand this aspect of life. They also provide a broader scope of possibilities for how men might respond to female sexuality: for instance, they might admire it, be transformed by it, or fear it. While I agree with the view that it is not healthy to conceptualize womanhood solely through sexuality or the reproductive cycle, nonetheless, these models should be at least part of the complete picture of what women might be. In short, it is understandable why Fāṭimah is idealized as a young, frail virgin, but it is not always helpful.

However, despite her frailty, Fāṭimah is **vengeful**. For instance, the Prophet says 'whoever angers Fāṭimah angers me, and whoever angers me angers God' (Bukhārī 1981, no. 3510). (This *ḥadīth* is found among both Sunnis and Shiʿis but is sometimes understood differently.) Some say that, due to her anger at how the caliphate had been taken from ʿAlī, God raised the city of Medina off its foundations, until the people begged her to stop her imprecations, lest they be destroyed. In the hereafter, she will also successfully petition vengeance for her son's enemies; the following narration is instructive since it ties together many themes:

> One day, the Prophet, peace be upon him and his family, came to Fāṭimah, peace
> be upon her, and she was sad, and so he said to her, "What has made you

sad, O my daughter?" She said, "O my father, I have remembered the plains of resurrection, and people standing naked on the Day of Resurrection".

And so he said, "O my daughter, this is indeed a tremendous day, but Jibrāʾīl has informed me that Allah, the Glorious and Mighty, says [that] the first for whom the earth will split open on the Day of Resurrection is me, then my father Ibrāhīm, then your husband ʿAlī ibn Abī Ṭālib. Then Allah will send Jibrāʾīl to you with seventy thousand angels, and they will build seven domes of light upon your grave. Then Isrāfīl will come to you with three garments of light, and he will stop before your head and call to you: 'O Fāṭimah bint Muḥammad, stand for the Resurrection," so you will stand, safe from your fear, your nakedness (ʿawrah) covered; and Isrāʾīl will present the garments to you, and you will wear them. Rūfāʾīl will accompany you with a highbred female camel of light—its halter of pearl, with a litter of gold atop it. And you will ride it, and Rūfāʾīl will lead it by its halter; and with you will be seventy thousand angels with banners of glorification (tasbīḥ) in their hands.

'And when the caravan hurries along with you, seventy thousand maidens of Paradise (ḥūrīs) will receive you, rejoicing at seeing you; in each of their hands will be a brazier of light, from which the perfume of incense (ʿūd) will radiate without any fire. Upon them will be crowns of jewels inlaid with emeralds, and they will hasten to your right side.

'And when they reach your grave, Maryam bint ʿImrān [the Virgin Mary] will meet you with heavenly maidens similar to what is with you, and she will greet you; and she and those with her will travel on your left side.

'Then, your mother Khadījah bint Khuwaylid, the first of the female believers in Allah and His Messenger, will meet you; and with her will be seventy thousand angels; in their hands will be flags of takbīr (magnifying Allah). And when they are near to meeting, Eve will meet you with seventy thousand ḥūrīs, and with her will be Āsiyah bint Muzāhim, and they and those with them will accompany you.

'And when you have reached the middle of the gathering [...] a voice will sound saying "Lower your gaze so Fāṭimah the daughter of Muḥammad, peace be upon him and his family, may pass".

'And no one will look at you on that day except Ibrāhīm, the Friend of the Merciful, and ʿAlī ibn Abī Ṭālib. And Adam will seek Eve and see her with your mother Khadījah in front of her; then you will be given a minbar of light. And the closest of women to you on your left will be Eve and Āsiyah. And when you climb the minbar, Jibrāʾīl will come to you and say, "O Fāṭimah, ask your request (ḥājah)," and you will say, "O Lord, show me Ḥasan and Ḥusayn".

'And they will come to you, and blood will be gushing forth from Ḥusayn's veins, and he will say "O Lord, grant me today my right against the one who oppressed me". The Almighty will become angry at that, and at his anger, Hellfire and all the angels all will become angry [...]. The killers of Ḥusayn—and their sons and grandsons—will be engulfed in flames, and they will say, "O Lord, we were not present with Ḥusayn," and Allah will say to the tongues of Hell, 'Take them with your flames with the blue-eyed and blackened faces, and take the enemies of the family of the Prophet (nawāṣib) and throw them in the deepest pit of Hell, because they were harsher against the supporters (awliyāʾ) of Ḥusayn than their fathers who fought Ḥusayn and killed him.'

'Then Jibrāʾīl will say, "O Fāṭimah, ask your request (ḥājah)," so she will say, "O Lord, my followers". "So Allah will say, 'They have been forgiven,' and you will say, 'O Lord, the followers of my two sons,' and Allah will say they have been forgiven, and you will say, 'O Lord, the Shiʾa of my Shiʾa (the followers of my followers) [...].' At that point the creation will wish they were followers of

Fāṭimah (*fāṭimiyyīn*) with you, their nakedness covered, the difficulties gone, the entry [into Paradise] eased for them [...]". (Kūfī 1410 AH, p. 445)

Just like her followers may lament alongside her in hope of intercession, they may also **imprecate** her enemies with the same salvific expectation.

Unlike **warrior goddesses**, Fāṭimah does not strike down her enemies herself; rather, she channels divine wrath. She is a **passive** locus of divine power, rather than active; miraculous events happen through or around Fāṭimah. This is a classic archetypal understanding of femininity—the male is the active principle and the feminine is the passive, such as the yin and the yang, the *animus* and the *anima*, or ʿAlī and Fāṭimah. However, Fāṭimah's passivity does not mean that she is uninvolved. Rather, it is through her passivity that she acts, just like the moon reflects the light of the sun but still profoundly influences the world:

> Repeatedly we find in the Hadith reports of Fāṭimah's absolute similarity with her father. It is narrated how, as she walked to the mosque to deliver her sermon, her gait was just like that of Muhammad. 'Fatimah is part of me' said the Prophet. Part of his substance; part of what he is and what he represents. 'Whoever angers her has angered me.' Here she is passive again, and crucially passive. Her task, her function as this *hadith* presents it is to *respond*. It is how she is affected by people's deeds that those deeds are judged as the Prophet would have judged them. We see how her passivity here is twofold, first in her assimilation of the Prophet's quality, and then how that quality as replicated in her must be affected to be effective representative of the message. This function of twofold passivity can be seen in the seventh Imam's description of Fāṭimah: 'a truth teller, a witness'. She sees what is done and describes it as it truly is [...]. Fāṭimah's role is to respond [...]. (Warner 2011, pp. 152–53)

Some Shiʿa today would argue against the portrayal of Fāṭimah as **passive**, saying that the sermon (*khuṭbah*) she delivered in defense of Fadak demonstrates that she took an **active** role in the community, modelling political activism for Shiʿi women (Feder 1955). In brief, the Prophet gave Fāṭimah a piece of land called Fadak. After he died, Abu Bakr seized it. She then went to the mosque in Medina and delivered a sermon in defense of her right to her inheritance. Twelver Shiʿi books contain a transmission of what is believed to have been her sermon, and, among Twelver Shiʿa, Fadak is not only about a piece of land but rather symbolizes the Imamate itself. That is, by usurping Fadak, Abu Bakr was usurping the caliphate; had he returned Fadak, he would have had to cede the caliphate to ʿAlī.

Historically, it is reasonable to describe Fāṭimah's stance as activist, both due to the Fadak sermon and due to other snatches recorded about her life; for instance, as outlined by ʿAlī Sharīatī (d. 1977) in his *Fāṭemeh Fāṭemeh Ast* (*Fāṭimah is Fāṭimah*). Sharīatī held that Shiʿism had become passive and ritualistic, thereby supporting the political status quo. He argued that this was not the true ethos of Shiʿism and that, instead, Shiʿa should rise up against unjust rulers (such as the Shah of Iran during his time) to work for social justice. While he applied this view equally to men and women, *Fāṭemeh Fāṭemeh Ast* attacks Shiʿi portrayals of Fāṭimah as passive, hidden, and helpless which he considered antiquated and instead rebrands Fāṭimah as a lifelong activist. Historically, his portrayal is sound; however, it clashes with the hagiographic ideals. Perhaps for that reason, despite the many changes in the world in the past fifty years, his critiques are still relevant to many Shiʿi women today.

Part of the reason why it is difficult to treat Fāṭimah's sermon about Fadak as a mere legal dispute in which Fāṭimah models how women can stand up for themselves is that, hagiographically, there is more at play. Fāṭimah begins and ends the sermon through lamenting; her lament is so powerful that, even before the sermon, 'all present began to weep'. The sermon is not pre-planned; rather, it is spontaneous oratory. Some see her eloquence a trait she inherited from her father; others present it as a model for female scholarship. However, we already know that Fāṭimah never needed to study or learn;

therefore, she is not actually modelling study. Rather, the sermon comes across as inspired, as if she is channeling God's words—and wrath:

> I have said what I had to say, being fully aware of your intention to forsake me and of the betrayal that has sprung in your hearts. But this was the unbosoming of the soul, the outburst of anger, the inability to further endure, the expression of the heart and the advancing of proof.

> So take its reins and saddle it [i.e., take Fadak], with its sore back and suppurating hooves, ever disgraceful, branded with the wrath of Allāh and eternal dishonor, leading to "the fire, set ablaze by Allāh, that roars over the hearts" (Q104:6-7), for what you are doing is witnessed by Allāh, "and they who act unjustly shall know to what final place of turning they shall turn back". (Q26:227) I am the daughter of "a warner unto you, before a severe chastisement" (Q34:46) So act, we too shall act, "and wait, we too shall wait". (Q11:122). (Jaffer 2023, p. 22; see also Clohessy 2009, pp. 155–58)

She concludes by turning towards her father's grave and reciting a poem mourning his passing. Similar characteristics are also found in the sermons attributed to Zaynab bint ʿAlī in the Battle of Karbala.

The reformist portrayal of Fāṭimah as a public activist also sits uneasily with the emphasis in the traditional account that, before delivering the sermon of Fadak, Fāṭimah put up and stood behind a curtain. Although heard, Fāṭimah remains **hidden** and **unseen**. This leads to the broader portrayal of Fāṭimah as hidden and unseen, well beyond the narrative of Fadak. While Shiʿi devotional art often portrays sacred figures, it is rare to come across a Shiʿi drawing of Fāṭimah which shows her face or form; rather, she is completely veiled (in cloth, in darkness, or in light). Even on the Resurrection, Fāṭimah will be raised clothed while others are raised naked, and angels will shield her from the gaze of others.[10] While, in practice, from childhood, the ṣaḥābah saw and recognized her, Shiʿi religious narrative emphasizes a narration which says that when ʿAlī and Fāṭimah married, the Prophet told ʿAlī to do the outdoor work and Fāṭimah to do the indoor work; Fāṭimah reigns over the **hearth** and **home**. Twelver Shiʿa also often assert that the Prophet asked Fāṭimah what the best thing is for a woman; in response, Fāṭimah told the Prophet that 'the best thing for a woman is for an unrelated man not to see her, and for her not to see an unrelated man', and the Prophet wholeheartedly approved. Regardless of whether or not she or the Prophet actually said this, the idealization of Fāṭimah—and hence women—as unseen is foundational to traditional Twelver Shiʿi religious culture: when women step forth in public in broad daylight (or, increasingly, on television or on social media), they are not quite acting like Fāṭimah.

This idealization of hiddenness can be taken two ways. On a mundane level, scholars such as Leila Ahmed argue that Muslim customs of women's seclusion and face veiling were rooted in pre-Islamic Mesopotamian custom, which shaped the nascent Muslim culture upon the expansion of the Arab-Muslim empire into Iraq. Since Iraq was pivotal to the development of Twelver Shiʿism, it is natural that local custom would have had a particularly strong influence. Second, the persecution experienced by the Shiʿa Abbasid caliphate may have led some Shiʿa to keep their female relatives out of the public sphere for their own safety; even some Shiʿi men operated under pseudonyms, and there is a strongly entrenched historical memory of the womenfolk of the Prophet's family being taken captive at the Battle of Karbala. On the most mundane level, it is simply taken as a sign that Fāṭimah exemplified sexual morality; that is, Fāṭimah teaches today's faithful women not to flirt.

However, in Islam, being unseen is also a trait of divinity. God is veiled through at least seventy veils; the upper heavens cannot be penetrated; and the Kaʿbah and Shiʿi sacred shrines feature coverings, drapes, and inner sanctums. The sacred is hidden, covered, and protected. The word *ḥaram* denotes both 'female relatives' and 'the sacred site of the Kaʿbah'. Even if *ḥaram* has devolved to 'harem' in orientalist discourse, nonetheless,

there is an underlying implication of something sacred about the feminine. While some trace this to pre-Islamic ceremony around the Kaʿbah (Brown 2019), it likely has archetypal roots, for instance, the womb as a dark, hidden space of creation. In narrations, Fāṭimah is likened to the 'Night of Power' (*laylat al-qadr*)—another symbol of **darkness**—which is an unknown night of great spiritual import, and drawings of Fāṭimah often depict her at night, sometimes with the **moon**.[11] Furthermore, Fāṭimah was said to have been named Fāṭimah because the creations were kept from (*fuṭima*, lit. 'weaned from') knowing her (Kūfī 1410 AH, p. 581). Fāṭimah is ultimately unknowable; disclosing her would be sacrilegious.

While these conceptions are profound in their implications for spirituality and womanhood, they do not translate well into everyday life. It is one thing to idealize Fāṭimah as an unknowable, unseen transcendental figure, and another to encourage normal women to go through life without contact with men or the broader society (even if this ideal has been shared by both Sunnis and Shiʿis). The breakdown of this ideal can be seen in post-war countries, where widows discouraged from public life have had to resort to begging or prostitution. It is telling that, in the Islamic tradition, an 'orphan' is a child who loses its father, not both father and mother; the woman is not expected to be able to provide. A fundamental theme of reform movements, Sunni and Shiʿi, has been to encourage women's participation in politics, education, finances, and the mosque. Sometimes this is advocated on the grounds that both men and women were involved in society during the time of the Prophet. Idealizing the invisibility of women without considering its ramifications in real life is both ahistorical and irresponsible. (This is especially the case for those who have never even lived in Muslim-majority cultures but celebrate it as akin to Marian chastity.)

In sum, the following feminine archetypes and images surrounding Fāṭimah in Twelver Shiʿi hadith, hagiography, and devotional art may be adduced:

- Transcendental, eschatological spirituality;
- Intercession;
- Hiddenness;
- Lamentation, suffering, and martyrdom;
- Vengeance;
- Darkness, the night, and the moon;
- Motherhood (life);
- Stillbirth (death);
- Marriage and domesticity;
- Youth and beauty;
- Passivity;
- The orphan;
- The damsel in distress;
- Virginity and asexualization;
- Injustice;
- The eternal sacred or the 'divine now'.

## 3. Other Sacred Women

Because Fāṭimah is understood to be sinless, divinely guided, and central to the universe itself, there is little impetus to look to other female role models, such as the wives or Companions (*ṣaḥābah*) of the Prophet. ʿĀʾishah, in particular, is not seen as a role model due to her stance against ʿAlī ibn Abī Ṭālib at the Battle of the Camel, and *ḥadīth* relate tension between her and Fāṭimah; this is in contrast to contemporary Muslims who present ʿĀʾishah as a role model of Islamic leadership and scholarship. Therefore, while some contemporary Sunnis embrace the female *ṣaḥābah* as models of healers, warriors, or scholars, in Shiʿism, the entire burden of womanhood is placed on Fāṭimah, as well as, to a lesser degree, her female descendants, such as Zaynab bint ʿAlī (d. 682) and Fāṭimah al-Maʿṣūmah (d. 816).

This is not to say that no other women are revered in Shiʿi sacred history. For instance, Shiʿis embrace a *ḥadīth* saying that the four perfect women throughout time are Āsiyah, the

wife of Pharaoh; the Virgin Mary; Khadījah, the first wife of the Prophet and mother of Fāṭimah, and Fāṭimah al-Zahrāʾ (Ḥanbal n.d., no. 2896). (A Sunni variant of this *ḥadīth* adds that ʿĀʾishah is superior to the four; it goes without saying that Shiʿis do not accept this version.) However, these four women do not traditionally present four different archetypes or a smorgasbord of options. Fāṭimah and the Virgin Mary, in particular, embody a virtually identical archetype (a **lamenting virgin mother intercessor**), except that the Shiʿi tradition emphasizes that Fāṭimah is spiritually superior. While Khadījah, the first wife of the Prophet Muḥammad and the mother of Fāṭimah, is also revered, traditionally, Shiʿis have not presented her as a role model for an independent woman, but rather focus on her status as the wife of the Prophet and mother of Fāṭimah, as well as a devoted Muslim. Additionally, while Sunnis hold that when Khadījah married the Prophet, she was in her 40s and had children from a previous marriage, Shiʿis often say that she was in her 20s—that is, she was **youthful** and **virgin**. Āsiyah, like Fāṭimah, is a **martyr**. Among later Shiʿi women, Fāṭimah al-Maʿṣūmah, the sister of the eighth Imam, is honoured, and many Shiʿis visit her shrine in Qum, Iran, seeking blessings or intercession. However, even though Fāṭimah al-Maʿṣūmah did not marry or bear children—that is, she is a **youthful virgin**—she is still treated as a **mother figure** who watches over local seminary students and is a caring **intercessor**; that is, she shares the same archetypal traits as Fāṭimah.[12] In contrast, a variety of feminine models—such as in a mythological pantheon—can 'help women appreciate the diversity between women and their various means of achieving fulfillment' and 'to search for new emotional and behavior alternatives' (Enns 1994, pp. 128–9).

## 4. Archetypes of Masculinity

A few words are also in order regarding archetypes epitomizing ʿAlī ibn Abī Ṭālib, her husband. Just as in Islam, masculinity and femininity are treated as opposites, so too are, in many ways, ʿAlī and Fāṭimah. While Fāṭimah is often portrayed in darkness, ʿAlī is often drawn in sunlight; an iconic portrait of ʿAlī shows him with a lion, symbolizing his epithet 'lion of God'. In some cultures, the **lion** and the **sun** symbolize masculinity, for instance, the association of the sun with Mithras. While Fāṭimah's ultimate authority is in the Resurrection, ʿAlī rules in this world. He holds the caliphate and the Imamate; he is statesman, scholar, warrior, and sage. While Fāṭimah is virginal, ʿAlī is virile;[13] while ʿAlī is visible, Fāṭimah is invisible. More obscurely, there are (rare) stories of ʿAlī slaying **dragons**.[14] While dragons or serpents are sometimes associated with goddesses, he, the masculine figure, is the anti-dragon; that is, the attribute is passed over to him antithetically. In short, the following attributes of masculinity are attributed to ʿAlī in Shiʿi hagiography:

- Earthly authority;
- Warrior;
- Statesman;
- Scholar, sage, and orator;
- Sun;
- Virility;
- Martyrdom;
- Dragon-slayer.

## 5. The Hand of ʿAbbās

The above list of archetypes and images associated with Fāṭimah al-Zahrāʾ encompasses many, although certainly not all, archetypes and images associated with goddesses in antiquity, and narrative archetypes in general. However, curiously, some of these missing archetypes appear in the figure of al-ʿAbbās ibn ʿAlī (d. 680), a revered martyr at the Battle of Karbala who was the son of ʿAlī but not Fāṭimah.

The main symbol of al-ʿAbbās in ritual iconography is the hand, sometimes referred to as the 'Hand of ʿAbbās'; it is associated with **power** and **protection**, and often appears on ritual flagpoles, on tapestries, and on water fountains. The 'Hand of ʿAbbās' relates to the same tradition as hand symbols throughout North Africa and the Middle East, primarily the

*khamsah* or 'Hand of Fāṭimah', which is primarily associated with women. (While people often envision the *khamsah* in the stylized form of three upright fingers and two lower fingers, it appears in many forms including a normal-proportioned hand.) The *khamsah*, in turn, is thought to have developed from imagery of the Carthaginian goddess Tanit, who was associated with fertility, war, youthfulness, the moon, water and sailors, lamentation, and stillbirth or child sacrifice. Given those feminine regional associations (Kemnitz 2022), the mere fact that the hand becomes the 'Hand of ʿAbbās' in Shi'ism is telling.

The main reason for the association of the **hand** with al-ʿAbbās is due to how al-ʿAbbās fell in the Battle of Karbala. The family of the Prophet and their supporters had been cut off from water by the enemy army for three days. Unable to bear the suffering of the thirsty children anymore, al-ʿAbbās asks his brother al-Ḥusayn for permission to ride out to the river and attempt to bring **water** to the children. An unmatchable warrior, he succeeds in collecting the water but is ambushed on the way back, and his hands are severed. Unable to defend himself, he dies.

Breaking down the imagery of the narrative, al-ʿAbbās is associated with **hands** and **water**; water is often associated with femininity and, in the Middle East and North Africa, the **hand** is as well, even if in other cultures it may also be associated with men. Although a great warrior, he is **obedient** to his brother and does not act independently. He is also the **protector of children**, especially girl-children, such as the young Sakīnah, who has a particularly close relationship with him. Unlike most of the other martyrs at the Battle of Karbala, al-ʿAbbās has a separate shrine which is like a sister shrine to that of al-Ḥusayn suggesting **pairedness**. He is also approached as an **intercessor**.

He is also referred to as the '**moon** of Banū Hāshim'; the moon and the sun have already been brought up. Gendering the moon in Shi'i imagery is difficult. Likely, Shi'i imagery reflects an overlay of masculine and feminine associations with the moon from different eras and cultures. There is a lack of agreement over whether, in pre-Islamic Arabia, the sun and moon were associated with masculine or feminine deities, or both: cultures in the Arabian Peninsula varied. Classical Islamicate astrology treated the moon as feminine and the sun as masculine. Karen Ruffle argues that the symbolism of the moon in Shi'ism in the Indian subcontinent reflects associations of the moon with Hindu gods, whereas Brown argues that associations of the moon with Hindu goddesses influenced pre-Islamic Arabia (Ruffle 2010; Brown 2019). Today, the moon does appear in drawings of Fāṭimah. In any case, in classical Shi'i hagiography, men and women—such as al-ʿAbbās and Zaynab bint ʿAlī—are both likened to the splendor of the full moon.

Altogether, this provides the following list of associations:

- Warrior;
- Water;
- Hands;
- Moon;
- Protection;
- Flag-bearer (war);
- Children (especially girl-children);
- Suffering (thirst, injury, empathy);
- Martyrdom;
- Sainthood;
- Obedience.

While al-ʿAbbās is unquestionably a male warrior archetype, some of these traits and associations are archetypally feminine, including more unexpected or obscure ones, such as the association with water. Therefore, certain feminine traits are sublimated onto him, making him, archetypally, the moon to Ḥusayn's sun.

## 6. Conclusions

Fāṭimah al-Zahrāʾ is the primary role model for Twelver Shi'i women; other sacred women in Shi'ism share her archetypal traits. Rather than merely being a historical figure,

she also has a transcendental, eschatological status; she is as near to divinity as a human can be in Islamic theology without violating the sanctity of God. Nevertheless, this does not immediately translate into female authority in the family, state, or religion. As has been wryly observed:

> [F]emale-based religion does not require matriarchy. This fact is clear to any visitor to places-for example, parts of India and Africa—where Goddess reigns supreme today in religion and the human female has little or no power in the workings of society. (Leeming and Page 1994, p. 23)

While reformist projects return to a historical portrayal of Fāṭimah to present her as a role model for female education (including seminary education), activism, and scholarship, historical and hagiographical notions of Fāṭimah clash. Archetypally, Fāṭimah has—as Jung would say—a 'magic authority' (Jung 1968, pp. 81–82). Her knowledge is intuitive and inborn; she never needs to study. Her sons are born divinely inspired and sinless, and therefore she cannot really model parenting. Other attempts become even more distant, for instance, presenting Fāṭimah as a model for female entrepreneurs or politicians. I am not in any way against this reformist movement—without it, I would not be writing this—and I wholly endorse women's education and political involvement, but attempting to treat her as historical and hagiographical (or worldly and otherworldly) simultaneously results in a paradox. At the same time, jettisoning the hagiographical involves jettisoning what makes Twelver Shi'ism what it is, as well as jettisoning the way humans seem to require these mythic archetypes to make sense of the human condition.

Archetypally, Fāṭimah embodies a classic understanding of the feminine in the dualist division of masculinity and femininity into active and passive, yin and yang, or *animus* and *anima.* She embodies numerous archetypes found cross-culturally and in antiquity, including Jung's light and dark mother figures; the young, innocent, virginal goddess; birth and death; and the lamenting goddess. She is the victim of injustice, rather than a bringer of justice. Although she transmits enormous spiritual power, she does this passively, through witnessing, responding, reflecting, or channeling. Conversely, her husband ʿAlī embodies qualities classically idealized as masculine, such as strength, authority, and virility. Absent from her portrayal are many feminine archetypes found in world mythologies, such as a crone, warrioress, huntress, or temptress. Some archetypes associated with femininity are sublimated onto al-ʿAbbās ibn ʿAlī. Portraying her as young, frail, innocent, and virginal omits a road map for later phases of womanhood, and does not mark out a path for women who are called to live out other archetypes, such as being an athlete or prime minister.

It has been observed that:

> Virtually all of the traditional archetypes applied to women—'the virgin,' 'the wife,' 'the lover or seductress,' 'the muse'—have as their basis sexual roles and expectations. That is, they are always defined in terms of their relationship to men. (Frontgia 1991, pp. 15–18)

This is especially true for Fāṭimah, as the ideal wife, the ideal mother, and the ideal daughter, and, cosmologically, as the divine mother figure. This persists in contemporary Shi'ism: women are still primarily celebrated as wives, daughters, and mothers. Furthermore, these archetypes are banalized: it is one thing to describe Fāṭimah as hidden because the sacred is hidden, and another to say that Fāṭimah teaches real-world women to keep out of the patriarchal male gaze.

Jung's 'mother figure' archetype has been criticized as assuming an inherent masculine–feminine duality, defanging women, and blaming the individual rather than society for institutionalized oppression. It:

> encourag[es] women to internalize oppression by attributing relational, nurturing skills to innate qualities rather than viewing them as survival mechanisms that help women find meaning in a world in which they hold lower social status. (Enns 1994, p. 128)

These criticisms equally apply to archetypal portrayals of Fāṭimah. For instance, portraying her as eternally lamenting and suffering sends the message that women are bound to suffer and lament. Belief that men and women are intrinsically different, and that men are active and public while women are passive and private, can be used to justify unequal social rights or the marginalization of women in the mosque, state, market, and court. Emulating Fāṭimah hagiographically can be impractical: most women do not only live as hidden, lamenting, passive intercessors who channel divine wrath (even if most women will confess to a few wrathful moments). Therefore, distinguishing between hagiography and history can help avoid pushing women to live as archetypes rather than living full lives.

Given the vast social change of the past century, how these archetypes are evolving or playing out in Iran—the world's most overwhelmingly Shi'i country—as well as among other Shi'a—is a complex and evolving question, not the least because Shi'a—like everyone else—negotiate a variety of cultural and religious influences. Shi'i cultures vary regionally, and Shi'a—like everyone else—vary individually both in their understandings of their faith as well as their own religiosity. While women's everyday lives have changed, and some fatwas have changed, Shi'i ritual narrative has been much more resistant to change, perhaps because it is based heavily on pre-modern texts such as *ḥadīth*, hagiography, and poetry, and is heavily intertwined with spirituality. While the reformist Fāṭimah or the reformist Zaynab may be discussed rationally, they appear much more archetypally in ritual storytelling and art. For instance, Shi'i commemorative ceremonies (*majālis*) are typically divided into two parts: a sermon, which is more sober and intellectual, and ritual storytelling (*rawḍah-khānī*), which is intended to evoke tears and is more ritualistic and emotional. The reformist Fāṭimah is more likely to appear in the first part, and the archetypal Fāṭimah in the latter. In any case, it is important to understand reformist ideas in their own context; even if contemporary Iranian Shi'i culture encourages stronger public involvement for women, women are not idealized as being 'out there' in front of men or as competing side-by-side with men in most arenas. Nonetheless, for nuanced explorations of how feminine archetypes are negotiated by both state and non-state Shi'i authorities in Iran, see Szanto (2021) and Shanneik (2023) (among many other writings).

Similarly, in recent years, further attention has been given to the question of female religious authority among Shi'a. In the seminary (*ḥawzah*) tradition, while women have been able to attend the *ḥawzah* and learn the teachings of Islam in order to instruct their families or other women, there is no popular archetype of a female scholar, nor are women envisioned as religious leaders. Rather, the universal image of a Shi'i scholar is a man with a turban, a father figure. Although, by now, slightly dated, Ziba Mir-Hosseini's observation on female seminary students in Qom remains worthy of consideration:

> After discussing the matter with male clerics and some female students, I concluded that in order to be accepted within a scholarly tradition as male-dominated and constructed as that of the Qom Houzeh, a woman must first observe its implicit rules. I found the same tendency in Cambridge among Old Girtonians, women of the first generation of female students in the University of Cambridge, who did not question many of the values of the Cambridge colleges but merely reproduced them in a different form. (Mir-Hosseini 2000, p. 18)

It is telling that the most recent book on female Shi'i authority (Kuenkler and Stewart 2021) contains contributions by women, Shi'is, and Iranians, but no actual contributions by Shi'i women who identify as scholars in the seminary tradition. Furthermore, challenges to patriarchal teachings or customs have only been broadly considered among Twelver Shi'a when they have been put forward by male scholars, such as Muḥammad Ḥusayn Faḍlullāh (d. 2010), who himself had many detractors; in Iran, many reforms have also been spurred by secular women's rights activists without seminary affiliations.

Jung's archetypes have been criticized for being rooted in white culture, for instance, in Greek mythology. Racially, this does not apply to Fāṭimah since she is non-white and from a part of the world that had closer ethnic, linguistic, and historical links to Africa. However, historically, Greco-Roman civilization left its mark on the Arabian Peninsula, and

the Greek philosophical tradition was revived during the Abbasid Empire. Therefore, this revives the question of where archetypes come from—are they inborn or cultural? In my view, archetypes ascribed to Fāṭimah reflect both sides. On the one hand, it seems almost certain that ancient mythological narrative archetypes were retained in the Middle East and North Africa after Islam and used to conceptualize Fāṭimah. (This seems particularly likely given the Shiʿi tradition of ritual lament.) At the same time, light and dark goddess figures are found in so many cultures that they likely reflect something innate to the human psyche or a foundational metaphysical reality. Jung's ideas have also been criticized as being biased towards the male or containing contra-sexual opposites of male and female; while those criticisms have merit, they do not necessarily apply here, since traditional Shiʿi hagiography shares both trends.

Heterosexual men often see women as mysterious, powerful, and frightening, and children often also idealize their mothers as a goddess figure—epitomizing beauty, wrath, nurturing, caring, and asexuality. However, this does not mean that this is who she actually is, outside her role of mom. Women, in contrast, tend to see each other in a more everyday light; there is nothing mysterious about cramps or catcalls. When men portray sacred women such as Fāṭimah al-Zahrāʾ, they sometimes are also navigating their own discomfort with women, and prefer to emphasize their asexuality and passivity to make them safe, familiar, and comfortable. Some men may also feel that elevating women to this goddess figure flatters or honours them, whereas some women may feel that this limits them. Since most Shiʿi portrayals of Fāṭimah—from the traditional to the reformist—have been developed by men, one wonders how women might independently have chosen to remember Fāṭimah, and how that would have impacted faithful women's daily lives.

## 7. Addendum

This paper was originally born from a study of the symbol of the hand (*khamsah*, today called the 'Hand of Fāṭimah'), which was presented by myself under the title of 'The Hand of Fāṭima (*khamsa/ḥamsa*) in Shiʿi Islam' at the Magic (Un)Disciplined conference, hosted by Societas Magica, at the University of South Carolina, 2022.

The symbol of a hand frequently used in Shiʿi ritual practice is often called the 'Hand of ʿAbbās'. Since the *khamsah* is said to have originated from imagery related to the Carthaginian goddess Tanit, and a stylization of a womb, I listed archetypes and images associated with Tanit and realized that almost all of them corresponded with archetypes and images associated with Fāṭimah al-Zahrāʾ: for instance, lamentation, stillbirth or child sacrifice, youth, beauty, and virginity. This is apart from a few—water, the moon, and hands—which, I realized, were sublimated onto al-ʿAbbās ibn ʿAlī, who shares the symbol of a hand. (At this point, someone observed that the image of a womb-hand atop a flagpole is not dissimilar to the cosmic image of creation involving a pen dipping into an inkwell, something I have not yet been able to unsee.) Furthermore, many archetypes absent from Tanit—such as the crone or temptress—are also absent from Fāṭimah al-Zahrāʾ.

I do not necessarily think that these archetypes came directly into Shiʿism from Tanit, although the presence of the Shiʿi Fatimid caliphate in North Africa could suggest some influence. Rather, I suspect these are regional ideas about feminine divinity shared across the Mediterranean and the Middle East in antiquity; for instance, in stories of Hera, Astarte, Neith, and Ishtar. Broadly speaking, they characterize what has been referred to, archaically, as the 'great Oriental goddess' (Frazer [1890] 1955, p. 404). I also suspect that they genuinely are universal archetypes which the human psyche is primed to respond to. Nonetheless, for those who are interested, Tanit is a path worth pursuing.

Lastly, I would like to thank Matt Melvin-Koushki and Marla Segol for their assistance at the conference as well as the other participants for their insightful observations. I would also like to thank George Warner for his comments on an earlier draft of this paper. I would also like to thank Rachel Woodlock for her encouragement and the willingness of her, Sariya Cheruvallil-Contractor, and Susan Carland to embrace a mythical approach to the question of lived experience.

**Funding:** This research received no external funding.

**Institutional Review Board Statement:** Not applicable.

**Informed Consent Statement:** Not applicable.

**Data Availability Statement:** No new data were created or analyzed in this study. Data sharing is not applicable to this article.

**Conflicts of Interest:** The author declares no conflicts of interest.

## Notes

[1]   Some of this paper specifically addresses Twelver Shi'ism, the largest contemporary Shi'i sect; other parts relate to Shi'ism broadly. I have attempted to be specific in my use of the terms 'Shi'ism' (in general) and 'Twelver Shi'ism' (specifically). The term 'sacred narrative' here denotes religious narrative intended to convey significant cosmic or spiritual truths and which is associated with ritual or doctrine. In the case of Shi'ism, this includes the use of sacred narrative in religious tracts, sermons, ritual storytelling, ritual theatre, devotional art, and religious film.

[2]   Along the same lines, one could consider Brown (2019) ('The Feminine Dimension in Islamic Esotericism') and Murata 1995 (Sachiko Murata, *The Tao of Islam: A Sourcebook on Gender Relationships in Islamic Thought*).

[3]   The seemingly paradoxical nature of this statement once impelled a mainstream publisher to make a last-minute 'correction' to a book on Islam, changing 'mother of her father' to the more logical 'daughter of her father'.

[4]   Not too much is known for certain about Manāt, although her name implies she was a goddess of fate and death, similar to other triad goddesses who wove fate, such as the Greek *moirai* (Buhl 1936).

[5]   Sulaym ibn Qays al-Hilālī is considered to be a pseudonym of a man who died around 678. Whether or not all the material traces to that time is disputed, the book nonetheless clearly contains early material and is one of the earliest, if not the earliest, collections of *ḥadīth*.

[6]   Historical accounts on how exactly ʿAbd Allāh ibn al-Ḥusayn was killed during the Battle of Karbala (680) differ; however, this is the story presented in Shi'i ritual storytelling, poetry, art, and theatre. On archetypal symbolism in the story of Karbala, see Inloes (2022).

[7]   Karen Ruffle, 'May Fatimah Gather Our Tears', p. 8. With respect to this theme regarding Ashura commemorations in general, see Ayoub (2011).

[8]   The movie was controversial primarily due to its unflattering portrayal of individuals revered by Sunnis. See Stolworthy (2022). Secondarily, certain casting choices can be noted, such as the selection of lighter-skinned actors for good characters and darker-skinned actors for evil characters.

[9]   On the Greek lament tradition, see Alexiou (1974).

[10]   See above narration quoted from *Tafsīr Furāt ibn Ibrāhīm*.

[11]   Corbin futher explores Ismaili gnostic ideas of Fāṭimah as *laylat al-qadr* in *Cyclical time and Ismaili Gnosis* as 'a Night which "is better than a thousand months", a Night in which "the Angels and the Spirit descend. . .. That night is a peace which endures until the rising of the dawn". This Night is said to be the typification of "our sovereign Fatima" (*mawlatuna Fatima*), daughter of the Prophet, mother of the line of Holy Imams, who, endowed with attributes similar to those of the Virgin Mother, gave birth to the succession of Epiphanies of celestial beings "until the rising of the dawn", that is to say, until the advent of the Perfect Child who will lead mankind back to its celestial archetype.' (Corbin 1983, p. 101). (The relationship between Fāṭimah and laylat al-qadr is further developed in Corbin 1986, p. 175.) However, this description does not fit wholly with Twelver Shi'ism. First, even though some authors such as Mary Thurlkill have likened the association of darkness with Fāṭimah to the darkness of the womb, birth is not typically symbolized by *laylat al-qadr* in Twelver Shi'ism. Second, in Twelver Shi'ism, the cycle of the births of the Imams stops at the Twelfth Imam (the Mahdī is believed to have been born in 869 CE, and to be waiting in occultation until the end times, at which time he will re-emerge, rather than be reborn) rather than continuing until the advent of the Perfect Child (the Mahdī).

[12]   For instance, see Majlisī (1403 AH, vol. 41, p. 317; vol. 60, p. 216; vol. 102, p. 266). Some time ago, a guidebook to her shrine was published in English under the awkward title *Qom and the Virgin of the City*, which in any case calls to mind the perception of her in the model of an intercessory virgin goddess.

[13]   For instance, he is said to have sired 28 children and married eight other women after the death of Fāṭimah.

[14]   See Ibn Husām's *Khāvaran-nāmeh* (15th c.). A page from a manuscript of this illustrating a dragon is housed in the Metropolitan Museum of Art (55.125.2). See also Shani (2017).

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
