# Peer review of "Ancient Feminine Archetypes in Shi‘i Islam"

_religions, doi:10.3390/rel15020149_

Round 1

Reviewer 1 Report

Comments and Suggestions for Authors

I'm a little surprised that the author did not discuss more of Henri Corbin's contribution to this subject, but over all it is a good article. The author may want to consider the life of Tahirih (Fatima Baraghani) who Babis regarded as the "return" of Fatima for a more activist interpretation of her life. Also, the Baraghani family operated a women's seminary in Qazvin which produced a lot of theological works written by women which might provide some different perspectives. Not that this article needs revision but it might provide some avenues for further research examining the views of Shi'ite women with clerical education. 

Author Response

Dear Reviewer,

Thank you for the ideas. I have added a paragraph on Henry (Henri) Corbin, a paragraph on Fatima Qurrat al-'Ayn, and a paragraph on the female seminary.

I think they are all good ideas to pursue, while at the same time, doing any of them justice would probably require a separate work.

Regarding Corbin, I prefer not to focus on him too much because he is focused extensively on ancient Persian culture specifically, and also more on Ismaili Shi'ism rather than Twelver Shi'ism. So not everything he says may relate directly to Twelver Shi'ism itself. Nonetheless he is an obvious choice to address especially when Jung is involved.

Regarding the hawzah, I would still argue that there is no archetypal notion of a female Shi’i scholar. Women who have attended the seminary are usually sidelined or ignored in practice, and women are not expected to have independent ideas. In order to be accepted in Shi’i scholarly circles, a woman needs to be ‘more Catholic than the Pope’ when it comes to patriarchy (i.e. more endorsing of patriarchal structures and merely repeating the views of men). Usually Shi'i women are better able to advance new ideas in academic circles. However, it is an interesting discussion, especially since the Shi'i hawza, like everything else, is continuing to evolve.

Best wishes!

Reviewer 2 Report

Comments and Suggestions for Authors

This is an excellent paper. It was enjoyable to read and the author should be commended for writing it. 

A few points to consider that may help improve it prior to final publication: 

- The introduction begins with (almost) a mea cupla in the form of a disclaimer. Academic work should not be compelled to justify itself to any forms of orthodoxy (religious or otherwise). As a Muslim, I can understand the difficulties in navigating faith and scholarship. However, recognising that human beings that exist in the world impact understandings of religion is not sacrilegious or blasphemous - rather, one may argue that recognising the difference between the divine and the human is in itself a religious quest. As such, I would suggest perhaps rethinking the introductory disclaimer and the apology at the end. You have nothing to apologise for. 

- This may be beyond the remit of this paper. However, it would be interesting to link the various conceptualisations of Fatima to the contexts that produce them. 

What about these contexts brings forth these particular usages of Fatima's symbolism? While you highlight the different tropes at play - these remain broadly descriptive, and as a reader, it would have been interesting to learn about why Fatima comes to be understood according to these archetypes. 

For example, the author mentions Shariati's Fatimeh Fatimeh Ast - and notes that Fatima is constructed as an activist. Shariati is, of course, a liberation theologian and attempting to foment revolution in pre-1979 Iran - his reading of the Islamic tradition is imbued with a strong revolutionary zeal - and therefore, his interpretation of Fatima equally reflects such. However, the paper does not provide any context to Shariati (or his conceptualisation of Fatima) or the other readings of Fatima. We are told what they are, and not why they may have come about. 

Doing so would provide a more accurate understanding of why Fatima is read in the way she is, and further to add to an understanding of why archetypes are invoked in place of which archetypes are invoked. 

This may not be the paper for that discussion - I leave that to the author's discretion - certainly something to consider for the future.

Author Response

Dear Reviewer,

Thank you for your close reading and feedback.

Disclaimer - I have revised the disclaimer and final paragraph to sound less apologetic. However, I left the actual "meat" of the paragraph because I think those issues are important to address directly (especially given the reality of sectarian violence). There is also a stylistic/literary function of the disclaimer paragraph, in that it foreshadows a few topics (like Manat) that come up later, so I'd like to keep the paragraph for that reason too.

Contexts - It is a good idea to follow up on. I've added a couple paragraphs about context where I think there are things that can be reasonably be said. However, I think that genuinely deducing the contexts of many of these ideas is quite complex, possibly in some cases historically unprovable; I would guess there have been many factors at play. 

Shariati - I added more discussion to better situate him. 

Many thanks!

Reviewer 3 Report

Comments and Suggestions for Authors

This article, which is how Fatima (a key female figure in Shiism particularly but also in Sunnism, too) is a major contribution to the field. The article provides important insights unto how Fatima figure is constructed as an ideal religious (gender) archetype. It is original. Besides, this is a comparatively difficult subject, as Fatima is a holly figure in Shiism. Equally importantly, the article contextualizes Fatima within the context of other architypes in various other religious traditions. The article incorporates relevant literature.

I strongly suggest its publication. Below, I list only several points that might help the author(s) in revising the manuscript:

In Abstract: “ Lastly, it helps to understand the hidden and unknown Fatimah.” This sounds too essentialist. This might be either removed or reframed.

Introduction: Introduction provides important points but, as it is, it seems a bit confusing. It might be better to reframe it by clearly demarcating the various issues mentioned in this section. Similarly, the sections where the author references Jung should be framed in a more systematic way. The author should also provide some other theories on idealizing personalities along with Jung. How is this subject treated in contemporary literature?

Lines 564-583. This concluding part of the section is highly compact. The list provided by the author is important. But one also needs further analytical evaluation of the preceding debates on Fatima.

Line 617: Why Ali is included into the debate. Given that he is “he” is it for a comparative reason or for he was Fatima’s husband. The beginning of this section requires some more theoretical articulations why we need to revisit Ali to understand better Fatima. This is also valid for the section that is on Abbas. Particularly the section on Abbas seems less relevant. The two sections (on Ali and Abbas) thus have the potential of diverting the focus and trajectory/course of the article. The author needs to rework on these two later sections. How are they relevant in terms of gendering architypes?

Conclusion may have some short insights un how idealizing Fatima relevant in contemporary Shiism particularly in Iran.

Author Response

Dear Reviewer,

Thank you for your close reading.

A number of revisions have been made to the article.

In some cases, I felt that adding more significant discussion might make it overlong, although they are good points to discuss. Here is a more detailed engagement with the points of the review:

Abstract - "hidden and unknown Fatima" - I deleted that. It had been an allusion to the hadith saying she is unknowable, but possibly some readers would not know that in advance.

Introduction - I revised it with the aim to make it clearer.

Other theories on idealizing personalities - There are already several put forward in the introduction and referred to throughout the article (not just Jung). I feel like this is sufficient for now. However, I am open to specific suggestions for theories or ideas that could generate new understandings. 

"How is this subject treated in contemporary literature?" - I wasn't entirely sure what "this subject" is specifically referring to (Jung? Fatima?). However, I added a mention of gendered critiques of Jung in the concluding section. (There is also an allusion to critiques of him at the beginning, but I felt that going in depth would go off topic; at the end of the day, the goal of the paper isn't to evaluate Jung.) 

Lines 564-583 - This wasn't actually intended to be a concluding section; rather, it introduces a new subject (hiddenness). I've added another sentence to signpost that to the reader. The tension between reformist ideologies and archetypes in hagiography is returned to in the conclusion (which has been expanded now) and is a general theme of the paper, so I think that works.

Abbas and Ali - In my view, the section on Abbas is very important here; it is major point that I have not ever heard anyone say before. That is, classically feminine symbols and ideas have been transferred to him (a man) in Shi'i hagiography.  So I think that is crucial to the idea of the negotiation of female archetypes. 

In revising the introduction, I tried to better call attention to the purpose of that. 

To a lesser degree, that happens with Ali, although that section is also there for comparison to highlight how masculine and feminine archetypes are situated against each other. 

Conclusion - Iran - I have added a paragraph on Iran. Of course, Iran is a very complex place, and doing it justice would require an entire book, at least! I don't think Fatima is un-idealized in contemporary Iran per se; there is just more of a blend of clashing ideas (ranging from secularism to reformist ideas to archetypal ideas).

Thank you again for the feedback. 

Best wishes!